# Lipid Peroxidation as a Possible Factor Affecting Bone Resorption in Obese Subjects—Preliminary Research

**DOI:** 10.3390/ijms241411629

**Published:** 2023-07-19

**Authors:** Marcin Maciejewski, Joanna Siódmiak, Bartłomiej Borkowski, Matthias Lorkowski, Dorota M. Olszewska-Słonina

**Affiliations:** 1Clinical Department of Orthopedics and Traumatology of the Musculoskeletal System, Antoni Jurasz University Hospital in Bydgoszcz, M. Curie-Skłodowskiej 9, 85-094 Bydgoszcz, Poland; jemako@wp.pl (M.M.); bartlomiejborkowski90@gmail.com (B.B.); matthiaslorkowski@gmail.com (M.L.); 2Department of Laboratory Diagnostics, Collegium Medicum, Nicolaus Copernicus University, M. Curie-Skłodowskiej 9, 85-094 Bydgoszcz, Poland; joanna.pollak@cm.umk.pl; 3Department of Pathobiochemistry and Clinical Chemistry, Collegium Medicum, Nicolaus Copernicus University, M. Curie-Skłodowskiej 9, 85-094 Bydgoszcz, Poland

**Keywords:** antioxidants, obesity, lipids, peroxidation, vitamin A, malondialdehyde, oxidative stress, bone turnover markers

## Abstract

Oxidative stress, which promotes bone catabolism, also affects the quality of bone tissue. We aimed to assess the impact of metabolic disorders and oxidant–antioxidant imbalance associated with primary obesity on bone resorption and formation processes. Anthropometric parameters, metabolic variables, oxidative stress indicators (malondialdehyde, vitamins A and E, uric acid, superoxide dismutase, catalase, glutathione peroxidase, type 1 paraoxonase, iron-reducing plasma antioxidant power) and markers of bone turnover (type I procollagen N-terminal propeptide and the type I collagen C-terminal cross-linked telopeptide; P1NP and CTX) were assessed in 108 Polish participants. Under the influence of oxidative stress, both enzymatic and non-enzymatic defense mechanisms were stimulated in obese subjects, especially in women, who had increased lipid peroxidation and activity of catalase (particularly in first-degree obesity) and decreased vitamin E concentration. The process of lipid peroxidation, as well as the weakening of the bone formation, was strongly manifested in women at a BMI range of 35.0–39.9 kg/m^2^ but not at BMI > 40.0 kg/m^2^, but it had a comprehensive negative impact on bone turnover in obese men. Obesity and its degree of advancement significantly affected the decrease in the concentration of the marker of bone formation—P1NP—only in the plasma of women. Excessive body weight had no effect on the value of the bone resorption marker in plasma, regardless of gender. Our results confirm the existence of the “obesity paradox” in the aspect of bone tissue metabolism and suggest that a specific body weight threshold changed the molecular response of the tissue.

## 1. Introduction

The available evidence suggests a complex and causal effect of obesity on bone health [1]. There are close relationships between adipose tissue and bone—in the bone marrow, both fat and bone cells arise from the same stem cells [2]. Although adipose tissue and bone appear to have fundamentally different functions, it is widely believed that excess adipose tissue has a protective effect on bone mass, based on clinical observations. The incidence of osteoporotic bone fractures, especially of the femoral neck [3,4,5], but not of the forearm [6], is lower in obese patients. However, fractures in obese or overweight patients account for a significant proportion of traumatic fractures occurring mainly in postmenopausal women or men over the age of 50 [7].

Under physiological conditions, reactive oxygen species (ROS) produced by osteoclasts can affect cell behavior as well as extracellular matrix composition and tissue architecture and stimulate and facilitate bone resorption, but their rising levels cause rapid bone loss, especially in postmenopausal women [8]. Excessive body weight exacerbates oxidative stress by increasing the amount of substrates for mitochondrial energy production and the excessive production of energy necessary to maintain additional amounts of tissue mass. Obesity is associated with an increased level of oxidative stress, accompanied by a decrease in the activity of both non-enzymatic and enzymatic antioxidants [9], promoting bone catabolism.

The concentration of markers of bone turnover, which are, among others, fragments of protein structural elements of bones or products of their degradation released into the circulation during the metabolic activity of osteoblasts and osteoclasts, is influenced by many factors. Their determination allows for the assessment of bone metabolism disorders. The International Osteoporosis Foundation and the International Federation of Clinical Chemistry recommend one bone formation marker (serum type I procollagen N-propeptide, s-PINP) and one bone resorption marker (type I collagen C-terminal cross-linked telopeptide, s-CTX) as reference markers for diagnosis [10,11,12].

The results of the research conducted so far on the relationship between body mass and the state of the skeleton lead to ambiguous and often contradictory conclusions, which do not include the interrelationships between the biochemical indicators of bone turnover and the parameters of oxidant–antioxidant balance. The aim of this study was to determine 1/the possible impact of obesity on bone turnover processes (the level of serum bone turnover markers—P1NP, CTX) and 2/to demonstrate the effect of disturbances in the oxidant–antioxidant balance accompanying obesity on the quality of bone tissue.

## 2. Results

### 2.1. Study Population

In the group of obese subjects, a higher percentage of women (70%) than men (30%) was noted. Basic information on the participants of the experiment is summarized in Table 1. In the group of obese subjects, cases of obesity, type 2 diabetes and hypertension were reported significantly more frequently in the family history than in the other participants of the study (81% vs. 18%, 54% vs. 8%, 70% vs. 0%, respectively), while coronary artery disease was reported less frequently (25% vs. 55%). Obese study participants less often declared practicing physical activity (12% vs. 27%).

### 2.2. Anthropometric Parameters

A significant, positive correlation between age and BMI was noted only in the group of participants without excessive body weight (rS = 0.38, *p* = 0.0135). Obese women (BMI ≥ 30 kg/m^2^) with WHR < 0.8 (gluteofemoral/gynoidal obesity) were significantly younger than those with androidal obesity (abdominal obesity; WHR ≥ 0.8) (28.5 ± 5.00 vs. 36.0 ± 13.0 years) (*p* = 0.0143). However, with the age of obese persons, the WHR value increased significantly (rS = 0.43, *p* = 0.0036). After taking into account the degree of obesity as a dividing criterion, the correlation between age and WHR values was demonstrated only in the population of women affected by third-degree obesity (rS = 0.54, *p* = 0.0201).

Among obese men (BMI ≥ 30 kg/m^2^), based on the WHR value, both gluteofemoral and abdominal obesity were found, while among women the androidal type of obesity was dominant. No age-related changes in the metabolic profile were observed in the group of patients. None of the anthropometric indicators of obesity affected the values of the parameters describing the carbohydrate and lipid metabolism of the patients.

### 2.3. Biochemical Parameters Describing Carbohydrate and Lipid Metabolism

The obesity of subjects who qualified for the study group did not affect their metabolic profile. The values of all biochemical parameters (total cholesterol—TC, fasting blood glucose—FBG, high-density plasma lipoprotein fraction—HDL, low-density plasma lipoprotein fraction—LDL, triglycerides—TG) of obese patients were within the ranges considered normal, but almost all (except LDL) significantly differed from those obtained in the control group (Table 2). There were no significant intergroup differences in the basic laboratory parameters describing fat and carbohydrate metabolism.

### 2.4. Oxidant–Antioxidant Status of Obese and Non-Obese Subjects

In the study group (BMI ≥ 30 kg/m^2^), significantly higher values of MDAp, FRAP and FRAP/UA were observed, while significantly lower values were observed for MDAe, GPx and vitamin E. The results suggest an increase in the level of lipid peroxidation, which may have been influenced by increased body weight and a decrease in the activity of factors playing a role in the antioxidant defense system—GPx and vitamin E (Figure 1).

In order to determine how weight gain affects changes in parameters describing the oxidative–antioxidant balance, the study participants were divided into subgroups using BMI as a division criterion (Table 3). The concentration of MDAp was significantly higher in people with abnormal BMI values, and in addition, this increase was dependent on the degree of obesity. The results obtained in the study group indicate a simple covariance between anthropometric indices (BMI) and the lipid peroxidation marker (MDAp) and, consequently, total plasma iron-reducing capacity (FRAP), which confirms the influence of obesity on the development of oxidative stress. As a result of the progressive destruction of membrane lipids, non-enzymatic defense mechanisms are also activated, which is reflected in the relationship between the concentrations of MDAp and vitamin E (rS = 0.29, *p* < 0.05) observed in the study group of women, strong enough to affect the image of the entire group of obese participants. The total FRAP level showed a significantly positive correlation (*p* < 0.05) with WHR (rS = 0.34), MDAe (rS = 0.29) and vitamin E (rS = 0.32) concentration.

The activity of SOD and PON1 was not dependent on gender, degree (BMI) or type of obesity (WHR). Excessive body weight, however, had an effect on the activity of CAT and GPx. The lowest mean CAT value was recorded in the group of non-obese men with normal WHR values (Figure 2). GPx activity was conditioned by the type of obesity—the average values of this parameter were significantly higher in people with WHR < 0.8/1.0 compared to study participants who did not show signs of visceral obesity.

The type of obesity was related to plasma vitamin E concentrations, but not vitamin A. These concentrations were significantly lower in the blood of people with visceral obesity.

After taking into account the division of the study population by sex, it was found that the total iron-reducing ability of obese women’s plasma depends on the concentration of vitamin E (rS = 0.40, *p* < 0.05) and uric acid (rS = 0.46, *p* < 0.05), while in the case of men it depends on the concentration of vitamin A (rS = 0.59, *p* < 0.05) and PON1 activity (rS = 0.45, *p* < 0.05).

Classification factors, such as gender and BMI, explain the variability in plasma iron-reducing capacity and FRAP/UA ratio but do not affect uric acid levels. The latter parameter is also not dependent on the type of obesity. It can be assumed that the plasma of obese study participants, especially men, has a greater ability to reduce iron ions than that of non-obese people. There were no significant differences in the values of parameters describing the oxidative status between obese subjects (regardless of the degree of obesity) declaring physical activity and patients belonging to the study group who did not declare any form of physical activity.

### 2.5. Biochemical Markers of Bone Turnover

Obesity only had an adverse effect on the concentration of the biomarker of bone formation (P1NP) and not on bone resorption (CTX) (Figure 3A). Taking into account gender as an additional dividing criterion, it was shown that excess body weight significantly reduced plasma P1NP concentrations only in obese women and not in men (Figure 3B).

Significant changes in the level of P1NP (the lowest value—41.5 ng/mL) were noted in people whose BMI was in the range of 35.0—39.9 kg/m^2^ (second-degree obesity) (Figure 3C). Interestingly, there were no significant changes in the value of this parameter in the plasma of subjects with third-degree obesity (Table 3). In the studied population, both obese and non-obese, no significant effect of physical activity on the concentration of bone turnover markers was found.

The inverse covariance between age and bone turnover markers is not surprising, although it manifests itself only in the population of obese study participants: P1NP (rS = −0.37, *p* = 0.0035) and CTX (rS = −0.28, *p* = 0.0305). Advanced age will have an adverse effect on the level of P1NP in both obese women (rS = −0.54, *p* = 0.0002) and men (rS = −0.50, *p* = 0.0304), as well as on the concentration of CTX, but this was only in women from the study group (rS = −0.35, *p* = 0.0171). The level of bone turnover markers in the non-obese study population was not related to age.

None of the parameters describing the carbohydrate and lipid metabolism of obese study participants affected the concentration of markers of bone formation and resorption in their plasma. However, it has been shown that the level of total cholesterol negatively correlates with the level of CTX in the serum of people who are not burdened with excessive body mass (CTX/cholesterol: rS = −0.33, *p* = 0.0367).

The higher the iron-reducing capacity of the plasma in the control group, the higher the P1NP level (FRAP/P1NP: rS = 0.33, *p* = 0.0359), but this was due to this relationship in healthy men (rS = 0.42, *p* = 0.0326) and not in women.

The concentration of P1NP is adversely affected by the increased level of uric acid, which has been reported only in the population of obese women (P1NP/UA: rS = −0.58, *p* = 0.0247). The only determinant of the oxidative–antioxidant status correlating with the concentrations of both bone turnover markers in the population of obese men was MDAp (P1NP/MDAp: rS = −0.48, *p* = 0.0338; CTX/MDAp: rS = −0.58, *p* = 0.0072).

## 3. Discussion

Adipose tissue is considered an endocrine organ that plays a key role in energy homeostasis. Various molecular pathways have been proposed through which adipose tissue communicates with bone. Both osteoblasts and adipocytes are derived from a common mesenchymal stem cell [2], agents that inhibit adipogenesis stimulate osteoblast differentiation [13,14] and, conversely, those that inhibit osteoblastogenesis increase adipogenesis [15].

Highlighting the pathophysiological relationship between obesity and bone metabolism is complex and continues to be an active area of research. Some of these indicate that abdominal obesity is associated with osteopenia and osteoporosis, while others suggest that the relationship is likely more complex and site-dependent, resulting in a lower risk for some types of fractures but a higher risk for others. In this context, scientists even mention the existence of the “obesity paradox” [16,17]. While body weight has a positive effect on bone formation, whether weight from obesity or excessive fat accumulation is beneficial for bone remains controversial.

Parallel to the main disorders concerning bone tissue metabolism, there are usually many other abnormalities, e.g., oxidative–antioxidant balance disorders, which have pathophysiological and clinical significance and could be useful for making preventive and therapeutic decisions. In the population of participants in this study, we demonstrated oxidative–antioxidant imbalances associated with obesity.

Among subjects who are overweight, researchers have identified a subgroup that lacks the metabolic abnormalities associated with obesity [18,19], also termed the “obesity paradox” [20,21]. It has been estimated that obesity with a normal metabolic profile occurs in approximately 20% of the obese population [22,23]. Such irregularities were not noted in our own research. The metabolic profile of the obese study participants was normal, although the values of the assessed biochemical parameters describing the carbohydrate and lipid metabolism (except LDL concentration) differed significantly from those obtained in the control group. This suggests that obese individuals, most of whom had a family history, require supervision and efforts to achieve lower LDL targets. Taking into account the different functional properties of various adipose tissue deposits, it is probable that the obesity of the subjects of our study group consists to a greater extent of subcutaneous and not visceral adipose tissue, which is confirmed by the lack of metabolic disorders. It can also be assumed that the state of maintaining the excessive body weight of the examined patients was not chronic.

Parhami et al. [24] showed that basal levels of cholesterol synthesis are necessary for osteoblast differentiation, but total cholesterol levels do not seem to directly affect the BMD of different populations [25,26,27]. The results of our own research confirm this suggestion in relation to obese people but contradict it when it comes to healthy participants. The CTX level, which reflects bone resorption and is often negatively associated with BMD [28,29], also showed this pattern in the non-obese population in our study. Reactive oxygen species, such as singlet oxygen and hydrogen peroxide, can induce the oxidation of cholesterol in lipoproteins and cell membranes. The oxidized sterols, including widely studied 7-ketocholesterol, can induce oxidative stress which inhibit osteoblast differentiation and finally lead these cells to a special form of death-oxiapoptophagy [30].

Despite the normal metabolic profile, the subjects declared disorders coexisting with obesity, and they concerned carbohydrate metabolism (13.6%), lipid metabolism (62.7%) and hypertension (42.4%). These disorders predispose obese individuals to the metabolic syndrome and osteoporosis [31,32,33]. The listed metabolic disorders can be treated as chronic diseases, initiating a series of stress reactions, which are not symptomatic due to the low age of the subjects (the median age was 36 years). However, the presence of excess adipose tissue in obese patients may serve as metabolic reserves during such hostile events, e.g., trauma or any other serious illness.

The skeletal system operates at different levels of metabolic activity in different disease states, which also change with age [34]. Most researchers believe that the age and gender of the subjects should be homogeneous in bone metabolism studies. Shao et al. [35] showed that the assessment of P1NP and β-CTX concentrations in the material from patients over 20 years of age does not require standardization of their age, and that the concentrations of these biomarkers did not differ significantly between sexes in the same age group. The authors suggest that a population need not be gender-homogeneous if its age is strictly matched, and vice versa. This cannot be confirmed by the results of our own research, in which significantly lower concentrations of only P1NP were obtained in the plasma of obese women compared to obese men belonging to the same age group. Kitareewan et al. [36] showed that in young adults the levels of bone turnover biomarkers are higher in men than in women, and then after the age of 30, they decrease (previously in women) and reach the lowest level in the fourth decade of life in women and in the fifth in men. Our results confirm the reports of Jørgensen et al. [37], and the marked decrease in bone turnover biomarkers probably reflects a change in the process of bone development and building to skeletal remodeling after reaching peak bone mass, which is more pronounced in obese women.

The levels of P1NP and CTX in obese patients differ in the published works. Some researchers have shown an increase in resorption [34] or both resorption and osteogenic markers [35]. In our studies, excessive body weight had no effect on the plasma CTX concentration in both obese women and men. The most obvious explanation, especially in men, seems to be coupled bone turnover: when inflammation (accompanied by obesity) impairs bone formation, the secondary effect observed will be impaired bone resorption, or vice versa [38]. The formation and resorption markers showed strong positive correlations with each other, which confirms this conclusion.

It is already known that food-induced suppression of bone turnover applies especially to bone resorption markers, but less is known about the response of bone formation markers [39]; therefore, the results obtained in the course of our research provide new information on primary obesity. Chronic diseases and inflammations, including obesity, are associated with bone loss and fragility fractures [40,41]. In the case of the studied population, it could also have an impact on the weakening of bone formation processes, expressed as a decrease in the concentration of P1NP in plasma.

The numerous reports published so far rarely specify the importance of the type of obesity in the context of lipid peroxidation [42,43]. In our studies, a significant decrease in MDAe concentration in obese subjects (almost 90% of the androidal type) and its increase in plasma in comparison to the control group were noted, which proves that the permeabilization of erythrocyte membranes occurs as a result of oxidative stress and confirms the existence of oxidant–antioxidant balance disorders, probably preceding the occurrence of metabolic disorders. These processes are most likely accompanied by hemorrhagic changes, such as the appearance of deformation of the erythrocyte membranes.

The process of lipid peroxidation, depending on the degree of obesity, was particularly strongly manifested in women with excessive body mass, as indicated by the correlation between BMI and MDAp recorded only in this study subgroup, but it negatively affected both the process of bone formation and resorption (stronger) only in obese men. Higher levels of lipid peroxidation biomarkers in women, especially older, may also be a result of a higher body fat content. Another reason for such an oxidative status in women may be the weakening of estrogenic potential antioxidant protection, which protects cells from oxidation and inhibits the generation of free radicals by neutrophils [44].

Although MDAp was the only parameter negatively correlated with plasma P1NP and CTX concentrations in obese men, it should be assumed that additional correlations will appear along with maintaining the state of obesity. Ozgocmen et al. [45] proposed that markers of oxidative stress such as erythrocyte CAT, SOD and GPx may be important markers of bone loss.

In our study group, as a result of the accumulation of high concentrations of ROS, the activity of CAT in erythrocytes increased (mostly in people suffering from first-degree obesity), adapting to such conditions. The results of studies using drugs that inhibit the formation of free radicals, for example, catalase, indicate a marked inhibition of bone loss [8].

In the studied population, under the influence of oxidative stress, both enzymatic and non-enzymatic defense mechanisms were stimulated, and their activation was more intense in the bodies of obese women. This scientific experiment showed a significant reduction in the concentration of vitamin E in the plasma of obese subjects compared to those not burdened with excessive body weight. What is more, the amplitude of the decrease in the level of vitamin E in the plasma of obese women was more than three times greater than in the case of men. Lowering the level of vitamin E destabilizes cell membranes, reducing their fluidity and elasticity. However, most indicators of the systemic antioxidant response remained unrelated to biochemical markers of bone turnover, which could have been influenced by either the short duration of excessive body weight or the moderate degree of oxidative stress.

The beneficial changes in the parameters of protection against oxidative stress observed in our research, expressed in the increase in the value of the plasma antioxidant potential (increase in FRAP), may indicate the efficiency of protective mechanisms or their increased regeneration in patients suffering from primary obesity, especially males. The highest antioxidant potential was found in the plasma of people with first-degree obesity.

Most of the in vitro and animal experiments on the interaction between oxidative stress and osteoporosis have yielded data that generally suggest a potential role for ROS in uncoupling bone turnover [8,45,46]. The weakening of bone formation processes (decrease in plasma P1NP concentration) was most evident at a BMI in the range of 35.0–39.9 kg/m^2^ (second-degree obesity) but not at BMI > 40.0 kg/m^2^. One would suspect that such an effect is the result of pharmacological treatment used in people with stage III obesity, but we know that such treatment has not been administered. Some of the study participants, both obese and of normal weight, used only dietary supplements.

Our experiment is a preliminary study and in order to confirm the existence of the “obesity paradox” in the aspect of bone tissue metabolism and to suggest the existence of a body weight threshold that changes the molecular response of the tissue, the range of determinations needs to be extended to include other biomarkers of bone turnover, e.g., N-terminal telopeptide of type 1 collagen (NTX), pyridinium crosslinks, total and bone specific alkaline phosphatase and osteocalcin. A limitation of the study is the small number of subgroups of participants selected on the basis of BMI, which we intend to take into account in the course of further research.

## 4. Materials and Methods

The study protocol was approved by the institutional review boards and the Bioethics Committee of the Ludwik Rydygier Collegium Medicum of Nicolaus Copernicus University (CM NCU) in Toruń (KB/477/2007 and KB/579/2007 with an annex, KB 36/2013 with an annex). The trial was performed in accordance with the principles of the Declaration of Helsinki and Good Clinical Practice guidelines.

### 4.1. Patients

Subjects living in the Kuyavian-Pomeranian Voivodeship with obesity of the 1st, 2nd and 3rd degree (defined by the body mass index (BMI)) [47,48], diagnosed and treated in 2013–2015 in the communal clinic Białe Błota, in the Department of Internal Diseases of the Multidisciplinary Municipal Hospital in Bydgoszcz, the Department of Family Physician CM NCU, in the Health Care Center “Communalni” in Bydgoszcz and in the Center for the Treatment of Obesity and Metabolic Disorders of the Gizińscy Medical Center in Bydgoszcz, qualified for this study.

In total, 108 people participated in this scientific experiment, including 59 women (mean age 36.1 ± 8.1 years) and 49 men (mean age 38.3 ± 11.2 years). The study group (mean age 38.6 ± 7.4 years) consisted of 47 women (mean age 36.2 ± 8.1 years) and 20 men (mean age 44.2 ± 13.0 years), whose BMI was within in the range of 32.6–57.6 kg/m^2^. Based on BMI values (greater than or equal to 30 kg/m^2^), three study subgroups were distinguished: obesity class I—BMI 30 to 34.9 kg/m^2^, obesity class II—BMI 35 to 39.9 kg/m^2^, obesity class III—BMI greater than or equal to 40 kg/m^2^.

The control (reference) group of 41 people, including 15 women (mean age 35.9 ± 7.4 years) and 26 men (mean age 34.0 ± 7.1 years), consisted of volunteers and blood donors reporting to the Regional Blood Donation Center and Blood Treatment in Bydgoszcz. The average age of normotensive subjects and those without obesity or metabolic disorders qualified to join the reference group was 36.7 ± 7.4 years. The BMI of people qualified to join the control group was within the normal range (18.5–24.9 kg/m^2^).

All participants in the study were volunteers and provided written informed consent before entering the study.

### 4.2. Research Material

Peripheral blood (plasma, erythrocytes and serum) was the material for testing the concentrations of lipid peroxidation products, uric acid, P1NP, CTX, vitamins E and A and the activity of antioxidant enzymes. Blood was collected from the cubital vein by qualified medical staff in the morning, from fasting patients (at least 8 h since the last light meal) in a sitting position, into glass tubes without the addition of anticoagulant and into tubes containing anticoagulant (K_2_EDTA, dipotassium ethylenediaminetetraacetic acid, BD Vacutainer, Medicalove, Gdynia, Poland). The serum was aliquoted into 2 mL Eppendorf tubes (MedLab, Raszyn, Poland), frozen at −20 °C and then −80 °C, and then stored until the determination of the parameters described below but not longer than 6 months.

The concentrations of lipid peroxidation products and the activity of antioxidant enzymes were determined in the biochemical laboratory of the Department of Medical Biology, Collegium Medicum in Bydgoszcz, Nicolaus Copernicus University in Toruń.

The concentrations of biochemical markers of bone turnover were determined in the Department of Laboratory Diagnostics of the Collegium Medicum in Bydgoszcz, Nicolaus Copernicus University in Toruń.

### 4.3. Assessment of Anthropometric and Basic Biochemical Parameters

In persons belonging to respective groups, waist and hip circumferences (to calculate the waist to hip ratio; WHR), height and weight (to calculate BMI; body mass index is the weight in kilograms divided by the square of the height in meters), as well as fasting blood glucose (FBG), total cholesterol (TC), high-density lipoprotein cholesterol (HDL-C), low-density lipoprotein cholesterol (LDL-C) and triglycerides (TG) were measured during routine examinations in diagnostic laboratories of health care units recruiting participants for the study. Hypertension was diagnosed based on the guidelines of the European Society of Cardiology (ESC) and the European Society of Hypertension (ESH) [49] for diabetes and dysplipidemia, based on The International Diabetes Federation (IDF) criteria [50].

### 4.4. Oxidative Stress Parameters

All measurements of the oxidative stress parameters were realized using different programs of the Varian spectrophotometer (UV/VIS Cary, Varian Inc., Palo Alto, CA, USA).

#### 4.4.1. Superoxide Dismutase (SOD) Activity in Erythrocytes

SOD activity in erythrocytes was determined by the method of Misra and Fridovich [51] which is based on the inhibition of the autoxidation of adrenaline by the enzyme to adrenochrome in an alkaline medium. SOD activity was measured at wavelength λ = 480 nm and expressed in U/g Hb.

#### 4.4.2. Catalase Activity (CAT) in Erythrocytes

The determination of CAT activity was performed by the method of Beers and Seizer [52]. CAT activity was measured during 0.0–0.5 min at wavelength λ = 240 nm and expressed in IU/g Hb.

#### 4.4.3. Glutathione Peroxidase (GPx) Activity in Erythrocytes

GPx activity in erythrocytes was determined using the method of Paglia and Valentine [53]. The absorbance was measured at wavelength λ = 340 nm between the second and fourth minute after the reaction was initiated.

#### 4.4.4. Paraoxonase 1 (PON1) Activity in Blood Serum

PON1 activity in blood serum was determined according to Playfer et al. [54] modified by Sogorb et al. [55]. The absorbance was measured every 1 min for 3 min, at room temperature, at a wavelength λ = 412 nm.

#### 4.4.5. Determination of Thiobarbituric Acid (TBA) Reactive Substances (TBARS) in Blood Cells and Plasma

Malondialdehyde (MDA) concentration in erythrocytes and plasma was determined according to Buege and Aust [56], modified by Esterbauer and Cheeseman [57], based on the reaction with TBA (Fluka, Sigma-Aldrich, Poznań, Poland) in an acidic environment (pH 2–3) and a temperature of 90–100 °C. The reaction results in the formation of a pink dye with an absorption maximum at the wavelength of λ = 532 nm.

#### 4.4.6. Ferric-Reducing Antioxidant Power of Plasma (FRAP)

Total plasma antioxidant capacity was determined using the method described by Benzie and Strain [58]. The method consists in measuring the increase in absorbance at the wavelength of 593 nm using a V550 Jasco TW 2.03 spectrophotometer (ELMI, Kraków, Poland) [59,60]. The activity of the tested compounds is expressed as the FRAP value (1 mM Fe(II) iron sulfate used as standard). A unit of FRAP is the ability to reduce 1 mM iron(III) to iron(II).

### 4.5. Concentrations of Vitamins A and E in Plasma

Vitamin A and E concentrations were determined using high-performance liquid chromatography (HPLC) [61]. The compound of vitamins was separated on a Kinetex 2.6 µm C18 75 × 4.6 mm chromatography column (Shim-pol, Izabelin, Poland) with a mobile phase composed of acetonitrile: methanol (95:5 ratio) (Merck, Taufkirchen, Germany), at a flow rate of 2 mL/min. Vitamin detection was performed with a UV/Vis detector at a wavelength of λ = 325 nm for vitamin A and λ = 295 nm for vitamin E. Vitamin concentrations were determined using Workstation Polaris software (USA) and expressed in µmol/L.

### 4.6. Uric acid (UA) Concentration in Blood Plasma

UA concentration, expressed in mg/dL, was assessed using POCT (point-of-care testing) diagnostics and the BeneCheck Plus system (General Life Biotechnology Co., Ltd., New Taipei City, Taiwan).

### 4.7. Bone Turnover Markers

#### 4.7.1. CTX (C-Terminal Cross-Linked Telopeptide of Type I Alpha Chain of Collagen)

Plasma CTX (bone resorption marker) concentration was determined using the IDS-iSYS CTX-I (CrossLaps^®^) Assay (Immunodiagnostic Systems Ltd., Boldon Business Park, UK) based on immunochemiluminescence (CLIA) technology for the quantification of CTX-I, a bone resorption marker, in human plasma on the IDS-iSYS automated multifunction system. The CTX-I concentration of each sample was automatically calculated and the reported concentration range was 0.033–6.000 ng/mL. Test sensitivity: limit of detection (LoD), 0.023 ng/mL; limit of quantitation (LoQ), 0.033 ng/mL.

#### 4.7.2. P1NP (N-Terminal Propeptide of Type I Procollagen)

P1NP plasma levels were determined using the IDS-iSYS Intact PINP test (Immunodiagnostic Systems Ltd., Boldon Business Park, UK), using the same method and apparatus as for the CTX marker. The P1NP (bone formation marker) concentration of each sample was automatically calculated and the reported concentration range was 2–230 ng/mL. The manufacturer’s reference values for healthy adults are 27.7–127.6 ng/mL. Test sensitivity: LoD, <1.0 ng/mL; LoQ, <1.0 ng/mL.

### 4.8. Statistical Analysis

Statistical analysis was performed using the Statistica 13.1 for Windows software (StatSoft, Poland). The normality of the results’ distribution was checked by the Kolmogorov–Smirnov test with the Lillefors correction. Research hypotheses were tested at the significance level α = 0.05. Qualitative data were analyzed using the χ2 test (χ2p Pearson and χ2Y Yates correction for low quantity, when observed or expected counts < 5). Due to the inconsistency of the distributions with the normal distribution of the analyzed variables, the results were presented as a measure of location; median (Me) and measures of variability: interquartile range (IQR) and minimum and maximum values in the described groups. The Mann–Whitey U test was used to test the significance of differences between the 2 groups in an independent design. The Kruskall–Wallis test was used to test the significance of the differences when comparing more than 2 groups in an independent design. The relationship between quantitative variables was analyzed using Spearman’s non-parametric rank correlation test, interpreted according to the J. Guilford scale.

## 5. Conclusions

The obesity of subjects belonging to the study group probably consists to a greater extent of subcutaneous adipose tissue, not visceral, and the condition of maintaining excessive body weight of patients was not chronic, which confirms the lack of metabolic disorders. In the studied population, under the influence of oxidative stress, both enzymatic and non-enzymatic defense mechanisms were stimulated, and their activation was more intense in the bodies of obese women. The increase in the concentration of MDAp of obese study participants confirmed the existence of disturbances in the oxidant–antioxidant balance in the course of primary obesity, probably preceding the occurrence of metabolic disorders. The lipid peroxidation, depending on the degree of obesity, was particularly strongly manifested in women with excessive body mass, but it negatively affected both the process of bone formation and resorption (stronger) only in obese men. Obesity and the degree of its advancement significantly affected the decrease in the concentration of the marker of bone formation—P1NP—in the plasma of women but not men. On the other hand, excessive body weight had no effect on the value of the bone resorption marker (CTX) in plasma, both in women and men. The weakening of the bone formation processes (decrease in the concentration of P1NP) was most visible at BMI in the range of 35.0–39.9 kg/m^2^ (second-degree obesity) but not at BMI > 40.0 kg/m^2^, which probably confirms the existence of the “paradox of obesity” in the aspect of bone tissue metabolism and suggests the existence of a body weight threshold that changes the molecular response of the tissue.

## Figures and Tables

**Figure 1 ijms-24-11629-f001:**
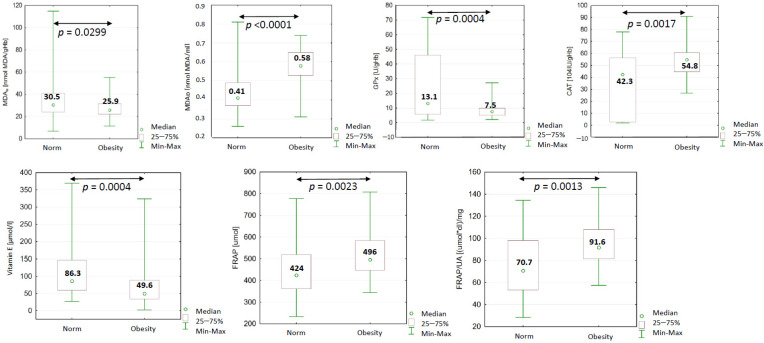
Values of blood oxidant–antioxidant balance indices of people with and without obesity (determined on the basis of BMI), significantly different in both groups (norm: N = 41; control: N = 67). *p*—level of statistical significance, MDAe—malondialdehyde concentration in erythrocytes, MDAp—plasma malondialdehyde concentration, CAT—catalase, GPx—glutathione peroxidase, FRAP—ferric-reducing antioxidant power of plasma, UA—uric acid.

**Figure 2 ijms-24-11629-f002:**
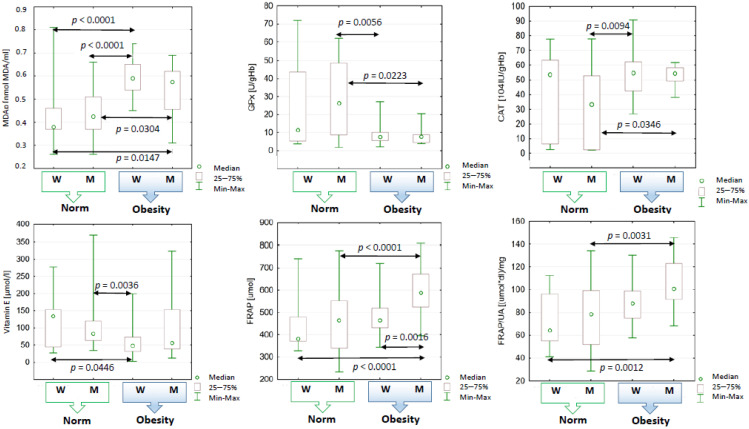
Statistical significance of differences regarding the values of blood oxidant–antioxidant balance indices of women and men with and without obesity (determined on the basis of BMI). W—women (norm: N = 15; obesity: N = 47), M—men (norm: N = 26; obesity: N = 20), Me—median, *p*—level of statistical significance, MDAp—plasma malondialdehyde concentration, GPx—glutathione peroxidase, CAT—catalase, FRAP—ferric-reducing antioxidant power of plasma, UA—uric acid.

**Figure 3 ijms-24-11629-f003:**
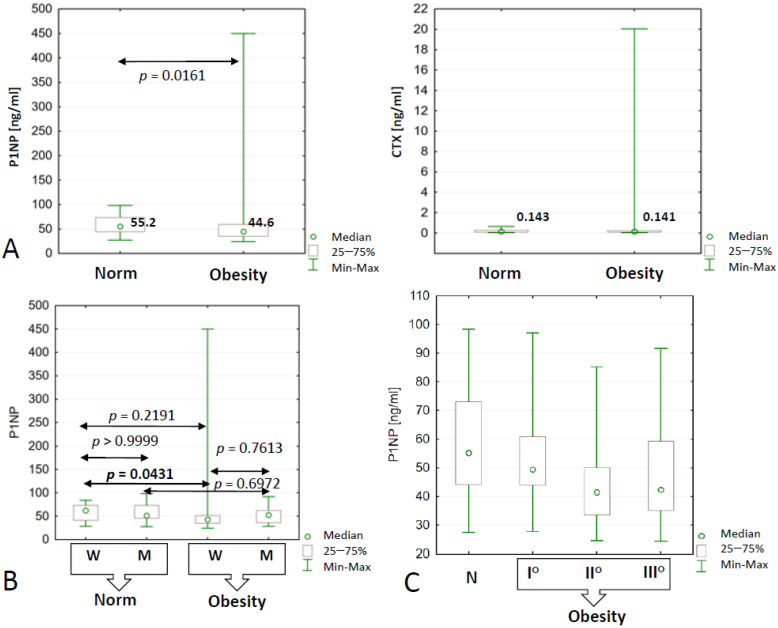
P1NP and CTX concentrations in the blood of obese (N = 67) and non-obese subjects (N = 41) (**A**). Significant *p*-values for multiple (two-sided) comparisons for P1NP; grouping variable: BMI and sex (**B**) as well as degree of obesity (**C**). P1NP—N-terminal propeptide of type I procollagen, CTX—C-terminal cross-linked telopeptide of the alpha chain of type I collagen, W—women, M—men.

**Table 1 ijms-24-11629-t001:** Characteristics of the control and study group.

			CONTROL GROUP BMI < 25 kg/m^2^ N = 41	STUDY GROUP BMI ≥ 30 kg/m^2^ N = 67	Statistics χ^2P^/^Y^χ^2Y^	*p*
Sex	Women		15 (36.59%)	47 (70.15%)	11.72	0.0006
	Men		26 (63.41%)	20 (29.85%)		
WHR	<0.8 (W) i < 1.0 (M)		36 (87.80%)	11 (16.42%)	52.73	<0.0001
	remaining		5 (12.20%)	56 (83.58%)		
Sex and type of obesity	W	gynoidal	-	42 (62.69%)	3.83	0.0526
		androidal		5 (7.46%)		
	M	gynoidal		14 (20.90%)		
		androidal		6 (8.96%)		
Waist circumference	Norm (W < 80 cm, M < 94 cm)		37 (90.24%)	1 (1.49%)	84.00Y	<0.0001
	Above the norm		4 (9.76%)	66 (98.51%)		
Medicines *	No		12 (75.00%)	41 (69.49%)	0.01Y	0.9047
	Yes		4 (25.00%)	18 (30.51%)		
Concomitant diseases	No		13 (76.47%)	41 (70.69%)	0.03Y	0.8731
	Yes		4 (23.53%)	17 (29.31%)		
Tobaccoism **	Non-smokers		17 (100%)	54 (83.08%)	3.32	0.1899
	Yes. up to 10/day		0	9 (13.85%)		
	Yes, up to 20/day		0	2 (3.08%)		
Obesity in the family	No		14 (82.35%)	12 (19.05%)	21.66Y	<0.0001
	Yes		3 (17.65%)	51 (80.95%)		
Diabetes	No		12 (92.31%)	52 (78.79%)	1.53	0.4646
	Yes		1 (7.69%)	9 (13.64%)		
	Impaired glucose tolerance		0	5 (7.58%)		
Diabetes in the family	No		12 (92.31%)	31 (46.27%)	7.52Y	0.0061
	Yes		1 (7.69%)	36 (53.73%)		
Hypertension	No		20 (90.91%)	33 (50.00%)	11.63	<0.0001
	Yes		2 (9.09%)	28 (42.42%)		
	125–140 mmHg/ 85–90 mmHg		0	5 (7.58%)		
Hypertension in the family	No		22 (100.00%)	20 (29.85%)	29.95Y	<0.0001
	Yes		0	47 (70.15%)		
Hypertriglyceride-mia	No		12 (92.31%)	25 (37.31%)	11.13Y	0.0003
	Yes		1 (7.69%)	42 (62.69%)		
Coronary artery disease in the family	No		12 (54.55%)	51 (76.12%)	3.73	0.0535
	Yes		10 (45.45%)	16 (23.88%)		
Declared physical activity	No		16 (72.73%)	59 (88.06%)	2.94	0.0866
	Yes		6 (27.27%)	8 (11.94%)		

BMI—body mass index (kg/m^2^); N—number; W—women; M—men; WHR—waist to hip ratio. *p*—test probability (χ2 test); *—dietary supplements; **—only tobacco smokers (excluding analogue smokers, e-cigarette users, people using nicotine patches or gums containing this alkaloid).

**Table 2 ijms-24-11629-t002:** Values of biochemical and anthropometric parameters in the control (C) and the study group (S).

Variable	BMI [kg/m^2^]	N	Me	IQR	Min	Max	*p*
Cholesterol [mg/dL]	C	41	178.0	35.6	153.6	342.4	0.0469
	S	67	193.3	51.1	155.2	279.0	
FBG [mg/dL]	C	41	84.0	12	68.0	112.5	0.0064
	S	67	96.0	11.9	82.0	108.0	
HDL [mg/dL]	C	41	92.0	35	40.0	131.3	<0.0001
	S	67	49.1	13	34.0	71.1	
LDL [mg/dL]	C	41	110.0	48.3	54.0	248.8	0.1649
	S	67	120.8	66.1	80.9	242.3	
TG [mg/dL]	C	41	59.0	21.2	41.0	254.8	<0.0001
	S	67	125.0	63.8	53.3	357.4	
WHR	C	41	0.860	0.109	0.657	1.000	<0.0001
	S	67	0.966	0.132	0.719	1.410	
Age [years]	C	41	34	7	23	56	0.0521
	S	67	36	15	23	77	

BMI—body mass index [kg/m^2^], C—control group, S—study group, FBG—fasting blood glucose, HDL—high-density plasma lipoprotein fraction, LDL—low-density plasma lipoprotein fraction, N—number, Me—median, IQR—interquartile range, TG—triglycerides, WHR—waist to hip ratio, *p*—statistical significance of differences.

**Table 3 ijms-24-11629-t003:** Values of oxidant–antioxidant balance indices and bone turnover markers in the blood of study participants after using BMI as a division criterion.

	BMI [kg/m^2^]	N	Me	IQR	Min.	Max.	*p*
Hb [mg/dL]	norm	41	16.93	3.56	12.59	18.91	0.5758
	obesity I°	18	16.50	1.73	14.27	19.09	
	obesity II°	25	15.85	1.40	13.98	18.64	
	obesity III°	24	16.36	1.20	13.40	19.02	
MDAe [nmol MDA/gHb]	Norm	41	30.48	17.06	6.83	115.0	0.1580
	obesity I°	18	25.48	8.32	17.27	49.1	
	obesity II°	25	24.63	8.26	13.21	52.8	
	obesity III°	24	26.52	9.46	11.52	55.1	
MDAp [nmol MDA/mL]	norm	41	0.41aBC	0.12	0.26	0.81	<0.0001
	obesity I°	18	0.5a	0.18	0.31	0.69	
	obesity II°	25	0.58B	0.08	0.31	0.74	
	obesity III°	24	0.60C	0.13	0.47	0.73	
SOD [U/gHb]	Norm	41	847.48	182.15	627.50	1492.51	0.8981
	obesity I°	18	894.40	87.81	597.11	1007.27	
	obesity II°	25	847.57	130.64	597.06	1036.99	
	obesity III°	24	831.24	237.71	590.59	1110.14	
GPx [U/gHb]	norm	41	13.10ab	40.22	1.73	71.76	0.0043
	obesity I°	18	6.95a	5.05	2.18	15.12	
	obesity II°	25	7.91	3.86	2.13	23.67	
	obesity III°	24	7.38b	4.27	3.84	27.14	
CAT [104 IU/gHb]	norm	41	42.30a	53.76	2.02	77.88	0.0132
	obesity I°	18	58.12a	12.92	36.44	80.39	
	obesity II°	25	53.36	15.26	26.89	90.83	
	obesity III°	24	54.73	14.43	34.61	69.86	
PON 1 [U/L]	norm	41	66.48	42.49	12.52	243.30	0.1580
	obesity I°	18	67.28	80.24	31.00	184.51	
	obesity II°	25	51.90	49.97	19.22	142.03	
	obesity III°	24	46.29	59.62	15.04	129.17	
Vitamin A [µmol/L]	norm	41	8.70	6.71	4.01	24.35	0.2608
	obesity I°	18	10.09	3.83	4.52	18.04	
	obesity II°	25	11.34	3.13	6.15	19.18	
	obesity III°	24	9.07	3.60	5.95	18.51	
Vitamin E [µmol/L]	norm	41	86.31ab	87.85	26.77	369.90	0.0041
	obesity I°	18	44.66a	54.00	22.09	199.69	
	obesity II°	25	49.15b	32.18	14.75	323.67	
	obesity III°	24	58.09	77.43	2.49	190.57	
FRAP [µmol]	norm	41	424.00a	159.00	234.00	777.00	0.0210
	obesity I°	18	508.00	96.00	344.00	720.00	
	obesity II°	25	496.00a	208.00	384.00	808.00	
	obesity III°	24	484.00	108.00	376.00	624.00	
UA [mg/dL]	norm	41	5.90	1.50	4.20	9.30	0.2811
	obesity I°	18	5.55	1.70	3.80	7.30	
	obesity II°	25	5.70	1.00	3.50	7.90	
	obesity III°	24	5.50	1.05	4.20	9.30	
FRAP/UA [(µmol*dL)/mg]	norm	41	70.67a	44.98	28.54	134.36	0.0102
	obesity I°	18	93.68	18.69	57.50	137.67	
	obesity II°	25	94.92a	24.52	62.95	145.88	
	obesity III°	24	88.18	23.92	68.28	130.29	
P1NP [ng/mL]	norm	41	55.20	28.9	27.40	98.40	0.0197
	obesity I°	18	49.45	17.0	27.80	97.00	
	obesity II°	25	41.50	16.6	24.60	85.20	
	obesity III°	23	42.30	24.2	24.40	91.70	
CTX [ng/mL]	norm	41	0.143	0.203	0.033	0.651	0.4411
	obesity I°	17	0.178	0.140	0.033	0.381	
	obesity II°	24	0.114	0.120	0.042	0.484	
	obesity III°	24	0.138	0.114	0.033	0.354	

N—number, Me—median, IQR—interquartile range, *p*—level of statistical significance (Kruskal–Wallis H test), Hb—hemoglobin concentration, MDAe—malondialdehyde concentration in erythrocytes, MDAp—plasma malondialdehyde concentration, SOD—superoxide dismutase, CAT—catalase, GPx—glutathione peroxidase, PON1—paraoxonase type 1, FRAP—ferric-reducing antioxidant power of plasma, UA—uric acid, P1NP—N-terminal propeptide of type I procollagen; CTX—C-terminal cross-linked telopeptide of the alpha chain of type I collagen.

## Data Availability

All data are contained within the manuscript. Source data are available in the Figshare repository (https://doi.org/10.6084/m9.figshare.21904470 (accessed on 27 June 2023)).

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
