# Peer review of "Lipid Peroxidation as a Possible Factor Affecting Bone Resorption in Obese Subjects—Preliminary Research"

_ijms, 2023, doi:10.3390/ijms241411629_

Round 1

Reviewer 1 Report

Maciejewski and colleagues conducted a study titled "Lipid peroxidation as a possible factor affecting bone resorption in obese subjects - preliminary research"

The study with a 70:30 female to male ratio is well written and ethically sound and approved by their respective committee, the methodology used can be replicated by future researchers.

Since the focus is an experimental or obese group, perhaps reporting and presentation should start with the obese group followed by the control and not visa versa.

The manuscript contains lot of acronyms, it would be ideal to provide list of abbreviations for readers to refer.

Table 1, include these two groups and their number in the table rather than the heading. As suggested, start with the experimental group followed by the control

Author Response

Dear Reviewer,

Thank you for your constructive comments that helped us improve our article. Done:

1/ Table 1 has been reformatted. The group sizes were left only in the table (removed from its heading), as recommended.

2/ A list of explanations of abbreviations used in the paper has been added, as suggested.

Reviewer 2 Report

 The incidence of oxidative stress on bone marrow resorption is not well known especially in obese subjects. The paper by Marcin M et al is clear and well designed and bring new arguments supportaing that oxidative stress can impact osteogenesis differently in obese women and men. There are strong argument on oxidative stress in the subjects studied but no explanation on potential mecchanisms is provided.

It is known that oxidative stress favor lipid peroxidation and that cholesterol oxidation occurs in the final process of lipid peroxidation. Cholesterol auto-oxidation mainly contributes to the formation of 7-ketocholesterol wich is known to have numerous cytotoxic effects on osteoblats. I think that it is important to introduce this hypothesis in the discusion and to site the following papers supporting this hypothesis.

* Ouyang J, Xiao Y, Ren Q, Huang J, Zhou Q, Zhang S, Li L, Shi W, Chen Z, Wu L.7-Ketocholesterol Induces Oxiapoptophagy and Inhibits Osteogenic Differentiation in MC3T3-E1 Cells. Cells. 2022 Sep 15;11(18):2882. doi: 10.3390/cells11182882. PMID: 36139457; PMCID: PMC9496706.

* Sul OJ, Li G, Kim JE, Kim ES, Choi HS. 7-ketocholesterol enhances autophagy via the ROS-TFEB signaling pathway in osteoclasts. J Nutr Biochem. 2021 Oct;96:108783. doi: 10.1016/j.jnutbio.2021.108783. Epub 2021 May 21. PMID: 34023424.

* Zarrouk A, Vejux A, Mackrill J, O'Callaghan Y, Hammami M, O'Brien N, Lizard G. Involvement of oxysterols in age-related diseases and ageing processes. Ageing Res Rev. 2014 Nov;18:148-62. doi: 10.1016/j.arr.2014.09.006. Epub 2014 Oct 14. PMID: 25305550.

* Anderson A, Campo A, Fulton E, Corwin A, Jerome WG 3rd, O'Connor MS. 7-Ketocholesterol in disease and aging. Redox Biol. 2020 Jan;29:101380. doi: 10.1016/j.redox.2019.101380. Epub 2019 Nov 14. PMID: 31926618; PMCID: PMC6926354.

* Sato Y, Ishihara N, Nagayama D, Saiki A, Tatsuno I. 7-ketocholesterol induces apoptosis of MC3T3-E1 cells associated with reactive oxygen species generation, endoplasmic reticulum stress and caspase-3/7 dependent pathway. Mol Genet Metab Rep. 2017 Jan 12;10:56-60. doi: 10.1016/j.ymgmr.2017.01.006. PMID: 28116245; PMCID: PMC5233792.

* Ouyang J, Xiao Y, Ren Q, Huang J, Zhou Q, Zhang S, Li L, Shi W, Chen Z, Wu L. 7-Ketocholesterol Induces Oxiapoptophagy and Inhibits Osteogenic Differentiation in MC3T3-E1 Cells. Cells. 2022 Sep 15;11(18):2882. doi: 10.3390/cells11182882. PMID: 36139457; PMCID: PMC9496706.

So, the discussion must be improve and the possibilty to use oxysterols as biomarkers for bone resorption in obsese subjects could be an interesting perspective

Author Response

Dear Reviewer,

Thank you for your constructive comments that helped us improve our article. Our study was a preliminary one, indicating the direction of further research. To date, we have only studied P1NP and CTX, but we plan to expand our studies to include other biomarkers and additional disorders in obese participants.

Done: We have supplemented the discussion with the suggested thread, concluding that it would be worth analyzing the relationship between oxysterols and the level of not only reference biomarkers of bone turnover in obese people. Thank you for pointing us to the literature sources, which we eagerly used.